# Attachment and the Development of Prosocial Behavior in Children and Adolescents: A Systematic Review

**DOI:** 10.3390/children9060874

**Published:** 2022-06-12

**Authors:** Mariana Costa Martins, Carolina Santos, Marília Fernandes, Manuela Veríssimo

**Affiliations:** 1William James Center for Research, ISPA-Instituto Universitário, 1149-041 Lisbon, Portugal; mariana.g.c.martins@hotmail.com (M.C.M.); mfernandes@ispa.pt (M.F.); 2Instituto Universitário de Lisboa (ISCTE), CIS-IUL, 1649-026 Lisbon, Portugal; acvss@iscte-iul.pt

**Keywords:** attachment, prosocial behavior, prosociality, empathy, childhood, adolescence

## Abstract

(1) Background: One key assumption of attachment theory is the relationship between security and the development of prosocial behavior. A secure child is more likely to feel and show concern for another individual, resulting in higher levels of prosocial behaviors (defined as voluntary behavior intended to benefit others—e.g., helping, sharing, comforting). (2) Method: Using a systematic review of the literature (PROSPERO: CRD42022290706), 703 articles were identified (EBSCO databases), from which 16 were considered eligible by the first two authors (inter-reviewer agreement: 85.714%). The criteria for an article’s exclusion were as follows: samples of children/teens not living in natural contexts; studies on psychopathologies; intervention programs; qualitative designs; studies on development or the validation of measures; studies that did not reliably measure the variables studied. (3) Results and Discussion: The eligible studies revealed incongruous results about the potential associations between attachment security to mothers and fathers and prosocial behavior. More consistent and significant relationships were found between the quality of attachment and empathy, while the associations between attachment and prosocial behavior were inconsistent (e.g., nine articles revealed significant associations; seven did not). In six studies, empathy was revealed to play an important role as the mediator between attachment security and prosocial behavior. The limitations and future recommendations were discussed.

## 1. Introduction

In recent decades, there has been a growing interest in the impact of attachment relationships on children’s social, emotional, and cognitive development. Bowlby [1,2,3] brought together the formulations of psychoanalysis, ethology, developmental psychology, and control systems theory to argue that an enduring, affective relationship with a caregiver promotes mental health and well-being throughout life [4,5]. First, it operates at a sensory-motor level and then moves to a more symbolic level during childhood, allowing the child to reflect and talk about the feelings of herself and others. During childhood, children actively construct their internal working models of attachment relationships [6]. In this way, attachment theory is established, at its core, as a theory of prosocial behavior [4].

One of the key concepts of attachment theory [1] is the existence of a caregiving system (from adult to child) that is fundamental to explain the existence and development of behaviors such as empathy, kindness, and care characteristics of sensitive interactions between adults and children. The caregiving behavioral system is inherently prosocial in nature, as it aims to relieve the distress of others. The caregiving system probably evolved due to its increased inclusiveness and adaptability, ensuring the survival and reproduction of family members [7,8,9,10,11]. This system is an entrance to the understanding of how prosocial behavior develops [4].

The caregiver behavioral system is wired to detect others’ needs and respond accordingly. However, it can be undermined by the caregiver’s anxiety and self-concern, so the quality of attachment is closely related to the effectiveness of the caregiver function. If security can promote empathy and prosocial behavior, insecurity can be related to self-concern, self-protection, and misjudged efforts to understand and help others [4].

Secure relationships present the child with a relational context where they can express and elaborate on their feelings, creating an optimal environment for the development of emotional understanding [12] that will promote prosocial behavior. Secure individuals are more comfortable with closeness, so they will probably be able to support and be more sympathetic to others [1,7,13]. Prosocial behavior is normally described in the literature as including social-emotional domains such as empathy, compassion, generosity, forgiveness, and altruism [7,14] and behavioral domains such as helping, sharing, and comforting [15].

Prosocial behavior is defined in the literature as voluntary behavior intended to benefit others [16]. It is central to group organization and for the establishment of cooperation between individuals [17]. For all stages of development, prosocial behavior is related to less loneliness [17,18], improved peer relationships and acceptance, and even school performance [19,20]. Theoretical explanations for the relationship between attachment security and a child’s capacity to care for others include variables such as self-esteem [21], empathy [21,22], and the social abilities of the child [23]. Others include components of the parent–child relationship such as positive parental affection [24].

### The Present Study

The question of whether individual differences in the prosocial behavior of children are related to parental attachment is still a key question in the literature due to the lack of consensus in the literature (some studies found significant differences in the child’s expression of emotions and prosocial behaviors that were associated with differences in attachment styles, while others did not; others found mixed results). Previous works reviewing this complex relation did not employ the methodology of a systematic review of the literature [4] or focused only on emotional dimensions and variables (e.g., sympathy, altruism), leaving a gap regarding the study and measurement of behavior, especially prosocial behavior [25]. For this reason, the main objective of this study was to implement a systematic review methodology in order to contribute to the literature on attachment and prosocial behavior [4,14].

## 2. Methods

### 2.1. Data Search Process and the Criteria for an Article’s Eligibility

The guidelines of the Preferred Reporting for Systematic Reviews were followed (PRISMA, [26]) in order to explore the relationship between attachment and prosocial behavior. Previously to any data extraction, the protocol of this review was registered on the International Prospective Register of Systematic Reviews, with the following PROSPERO number: CRD42022290706. 

A systematic searching process of the data was carried out using all of the EBSCO databases (e.g., PsycINFO, Psychology and Behavioral Sciences Collection). The following Boolean terms were entered: AB attachment AND (AB prosocial behavior OR AB prosociality). The combination of these terms was searched in the title, abstract, and keywords. The search was applied until 15 February 2022 and resulted in 703 records. No timeline restrictions were imposed during this initial search procedure, seeing as how recent the resulting articles were from the start (the oldest was from the 1980s).

First, the screening of the articles’ titles was conducted, where duplicates were cleared out and the selection and exportation of the relevant studies were performed, using a priorly established list of inclusion and exclusion criteria (see also, Table 1). The list of inclusion criteria included: (1) empirical research with an available abstract published in peer-review journals; (2) studies that were in Portuguese, English, French, Italian, or Spanish (languages mastered by the authors); (3) studies analyzing the associations between parental attachment and prosocial behavior. The abstracts were screened by the first and second authors to assess whether the paper was eligible and met these criteria. Those that did not meet the criteria were removed. Disagreements and discrepancies were always discussed until a consensus was reached. If a consensus was not achieved, two other independent reviewers were consulted. Finally, the full texts of the remaining articles (the ones selected through the abstract screening) were read and screened, and the same inclusion and exclusion criteria and selection process were used. 

The criteria used for the exclusion of papers included (see Table 1): (1) participants living in non-natural environments (e.g., institutions); (2) studies on attachment or prosocial behaviors within the context of psychopathologies (e.g., substance abuse); (3) studies on intervention programs; (4) papers mainly aiming to validate measures; (5) studies with qualitative designs; (6) non-peer-reviewed papers (e.g., books, chapters, conferences, posters); (7) studies that used instruments that did not follow Bowlby’s or Ainsworth’s conceptualization to measure attachment. 

### 2.2. Study Selection Plan

A total of 703 articles were initially obtained through the databases and were screened by the first author, following the established and previously mentioned inclusion criteria and resulting in 671 articles being excluded. The abstracts of the remaining 32 articles were screened by the first and second author to determine if they were eligible and followed the inclusion criteria; only 21 were selected, and the respective full texts were further assessed independently by the first two authors for inclusion and eligibility. Finally, 16 articles (listed in Appendix A) met all the inclusion criteria and were deemed eligible (Figure 1). Discrepancies were always discussed until a consensus was reached. 

All of the steps and procedures of this systematic review (identification, screening, and selection of eligible studies) are synthesized in Figure 1, as previously detailed.

### 2.3. Data Extraction Plan

The data extraction was carried out by three reviewers. Categories were established to summarize the results of the 16 selected studies and with the intent to identify (1) the overall characteristics of the studies (i.e., country of origin and theoretical background); (2) the overall characteristics of the samples used (i.e., socioeconomic status and age); and, finally, the (3) assessments of prosocial behavior (see Table 2, Results). This categorization of the retrieved articles was mainly conducted by the first author; however, the remaining reviewers were always consulted during this process. All disagreements or discrepancies were discussed until a consensus was reached.

The validity and quality of the studies were assessed through the Quality of Survey Studies in Psychology Score (Q-SSP, created by the OSF from the Center for Open Science; see https://osf.io/5aepd/, accessed on 1 May 2022), the most adequate index for the various designs and instruments (e.g., questionnaires, observational measures, and scales) that are used in empirical psychological research.

## 3. Results

### 3.1. Theoretical and Empirical Perspectives

As a theoretical background, the eligible papers also resorted to social psychology but mainly referenced developmental psychology (for example, the attachment theory and the framework on social-emotional development, while citing and referencing relevant authors on both topics, such as Bowlby, Cassidy, Asher, Waters, and Eisenberg). For the present review, studies that used secondary data were not found (Table 2). Merely five studies revealed a longitudinal design (31.25%). The majority of the studies used child/adolescent-reported measures to assess both attachment and prosocial behavior, while a minority used observational measures to assess prosocial behavior (11.76%) and parent-reported instruments to assess attachment (also 11.76%; see Table 2).

**Table 2 children-09-00874-t002:** Categorization and description of the eligible studies and respective samples.

Studies Descriptives	Total of Articles (*n*)	Percentage (%)	Article ID ^a^
*Theoretical background:*			
Developmental psychology (socio-emotional development)	14	73.68%	1–10, 12, 14–16
Social psychology	5	26.32%	6, 10, 11, 13, 14
*Type of data:*			
Original	16	100%	1–16
Secondary	0	0%	-
*Study design* ^b^			
Longitudinal	5	31.25%	2, 3, 4, 6, 15
Cross-sectional	11	68.75%	1, 5, 7–9, 10–14, 16
*Assessment of prosocial behavior*			
Child/Adolescent-reported	7	41.18%	5, 6, 11, 13, 14, 15, 16
Parent-reported	4	23.53%	4, 7, 8, 12
Teacher-reported	4	23.53%	2, 3, 10, 12
Observation	2	11.76%	1, 9
*Assessment of attachment*			
Child/Teen-reported	11	64.71%	3, 5, 6, 8, 10–16
Parent-reported	2	11.76%	6, 7
Observation	4	23.53%	1, 2, 4, 9
**Samples Characteristics**	* **N** *	**%**	**Article ID ^a^**
*Country of origin*			
North America	7	43.75%	1, 2, 4, 5, 7, 9, 12
Europe	4	25%	3, 8, 14, 15
Oceania	1	6.25%	13
Africa	1	6.25%	10
Asia	3	18.75%	6, 11, 16
*Age group*			
Children	10	58.82%	1–4, 7–11, 14
Adolescents	7	41.18%	5, 6, 11–13, 15, 16
*Socioeconomic status*			
High/Moderate	8	42.11%	2–4, 7, 11–13, 15
Low	4	21.05%	1, 3, 4, 13
Not mentioned	7	36.84%	5, 6, 8, 9, 10, 14, 16
**Assessment of Prosocial Behaviors**	* **N** *	**%**	**Article ID ^a^**
Global score	16	84.21%	1–16
Helping	1	5.26%	1
Sharing	1	5.26%	1
Comforting	1	5.26%	1

a. Article references are presented in Appendix A. b. According to the inclusion criteria of the current review, only the quantitative results of studies employing mixed methods were included. Note: some categories (e.g., theoretical background, assessment of prosocial behavior) are not mutually exclusive.

### 3.2. Samples and Assessments

The majority of the studies were from North America (more specifically, 43.75% in the USA) or Europe (25%). Studies on children (58.82%) were slightly more predominant than research involving adolescents. Most samples predominately presented participants of a medium-high economic status (42.11%), although a significant proportion of the authors did not assess or mention the socioeconomic status of their participants (36.84%; Table 2). All of the studies unanimously chose to approach the assessment of prosocial behaviors with a global and final score; however, Beier and colleagues [14] added an individual assessment of behaviors such as helping, sharing, and comforting.

The bulk of the participants were predominately Caucasian. The most frequently used instrument to measure adolescents’ parental attachment was the Inventory of Parent and Peer Attachment (IPPA, [27]), a questionnaire adapted to different languages (e.g., Spanish, Chinese), and the most consistent instrument used to measure attachment during childhood was the Attachment Q-Set [28,29]. The Strengths and Difficulties Questionnaire (SDQ, [30]), a self-reported questionnaire, was the most commonly chosen instrument to measure prosocial behaviors among older children and adolescents. Observational measures were preferred to assess prosocial behavior in studies involving younger children.

Various research designs and statistical approaches were taken in the different studies. The most common approach was testing specific conceptual models, i.e., [14,22,24,31,32,33,34], where empathy played a frequent and significant role as a mediator between attachment security and prosociality. Secondly, we also frequently found Pearson’s correlations in the extracted results, i.e., [21,35,36,37,38,39]. Despite taking the same approaches, the results were incongruous with each other. Further individual assessments of the participants, instruments, and results of the selected articles are presented in Table 3.

In summary, Beier and colleagues [14] revealed a robust positive association between attachment security and children’s spontaneous prosocial and helping behaviors. Bureau and Moss [40], in contrast to Eceiza and colleagues [23], found no significant differences in prosocial behavior levels considering different attachment styles (secure, ambivalent, avoidant, and disorganized). However, these authors revealed that children with a disorganized attachment classification or representation developed higher externalizing scores than secure and avoidant children.

Kim and Koschanka [22] showed that for mother- and father-child dyads, security moderated the path from empathy to prosociality. Insecure and unempathetic children were particularly low in terms of prosociality.

In contrast to Panfile and Laible [36], Profe and colleagues [32], and Simons and colleagues [58], studies such as those by Laible and collaborators [21], Shoshani and collaborators [37], and Thompson and Gullone [38] found significant and positive correlations between adolescents’ quality of attachment and prosocial behaviors. Tur-Porcar and colleagues [39] took it even further and found positive and significant correlations between prosocial behaviors in children and attachment to both parental figures, i.e., mothers and fathers. Marcus and Kramer [35], in turn, demonstrated how prosocial initiative and orientation are positively and significantly correlated with attachment security and negatively and significantly correlated with attachment insecurity.

Laible and colleagues [21], who studied adolescents, also pointed out the potential role of prosocial behavior as a mediator between parental attachment and self-reported self-esteem. Li and collaborators [31] provided evidence for their conceptual model and showed how attachment security is positively and significantly associated with prosocial behavior, as opposed to attachment ambivalence.

Predictive multiple regression models also showed incongruous results (attachment as a significant predictor of prosociality in children: Beier and colleagues [14]; non-significant: Michiels and collaborators [24]). Regarding adolescents, the structural equation models of Zhao and colleagues [34] showed non-significant associations between these two study variables, i.e., [32,34].

## 4. Discussion

The present systematic literature review revealed some inconsistency in the results reported by different studies, which is in agreement with what had been previously reported by authors such as Shaver [4] and Beier [14]. Even studies with corresponding quality, designs, and statistical tests reached different conclusions. For example, Bureau and Moss [40] found no significant differences in the levels of prosocial behaviors across the different attachment styles, but Eceiza and his collaborators [23] recorded higher values of prosocial behaviors with the secure and ambivalent styles (when compared to the avoidant style). Despite this inconsistency, it is important to stress that 11 out of the 16 selected papers revealed a significant association between the two domains under consideration. These significant associations are in line with what has always been advocated by attachment theory, i.e., that caring and responsive parental and attachment figures promote secure internal models that allow the child (or adolescent) to regulate his or her emotions and to be able to care for others [1,2,3,4,14]. This theoretical framework is in line with the empirical findings of the selected studies (which predominantly found positive associations between the variables being studied). It is also consistent with the fact that only one of the articles, by Simons and colleagues [58], found a negative association between parental attachment and prosocial behaviors. However, even taking this result into account, Simons [58] did not find a significant association (see Table 3). Briefly, attachment to one’s mother and attachment to one’s father revealed similar evident associations with prosocial behaviors in children, e.g., [39].

The complexity with which attachment and prosocial behavior relate and develop is noticeable in the conceptual models and designs using structural equation models in the selected studies. In these models, a significant role of empathy as a mediating variable stands out, i.e., [21,22,32,36,38].

It should be added that many of the selected studies were of good quality (Table 3, Q-SSP cut-off point: 13), but some were only marginally good or of threshold quality (a score of 10 or above, except for one study by Marcus and Krammer [35]. This indicates the absence of empirically relevant information in the studies presented here—particularly, information needed in a psychology research context.

The samples presented in this review, by the current literature, were revealed to be skewed and lacking in regard to cultural and social diversity (noticeably, half of the selected studies presented predominantly Caucasian samples (Table 3) and a medium-high social status (Table 2)). This represents a gap in the literature on the reporting and understanding of the different contexts and resulting social nuances.

Only sixteen papers were considered and extracted, and the present systematic review clearly indicates that this is a topic that needs further empirical exploration. Several questions remain. For example, besides empathy, what are the other possible mediators between attachment and prosocial behavior (e.g., control variables such as verbal intelligence; sociodemographic and emotional variables)? Further, biological (e.g., the presence of relevant hormones such as cortisol or oxytocin) and contextual variables (social environments and ideologies) were either not measured or not highlighted in the results found.

Another important goal is to expand and develop the definition of prosocial behaviors that can be observed. To date, only Beier [14] referred to helping, sharing, and comforting but did not clarify how each behavior can be associated with each developmental stage. Certainly, these behaviors can have different dynamics throughout childhood, puberty, and adolescence and can be differently related to peer interactions and friendships. Additionally, research should take into consideration the differential role of facilitating and non-facilitating (social) contexts, especially if prosocial behavior is different in function of the target. Future studies should elect a longitudinal design, explore different mediating variables, and, if possible, use observational measures. Finally, it is fundamental to address the possible differentiating contributions of paternal and maternal attachment.

## Figures and Tables

**Figure 1 children-09-00874-f001:**
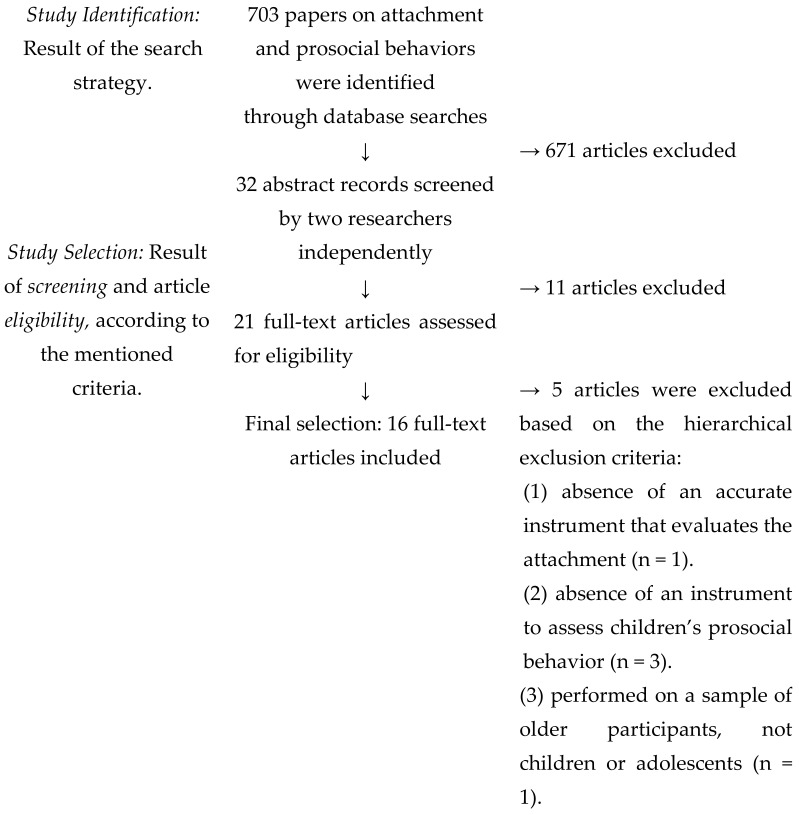
Flowchart of the full process of the identification and selection of the studies (according to the PRISMA, Page et al., 2020 guidelines).

**Table 1 children-09-00874-t001:** Complete list of the inclusion and exclusion criteria followed.

Inclusion Criteria	Exclusion Criteria
(1) Empirical research published in peer-reviewed journals with an available abstract; (2) Papers written and published in Portuguese, English, French, Spanish, or Italian (languages mastered by the authors); (3) Studies analyzing the associations between parental attachment and prosocial behavior. (4) Research on children and adolescents (with ages ranging from 0 to 19 years old).	(1) Papers with samples of children or adolescents in non-natural contexts (e.g., institutions; focus on the current pandemic context); (2) Studies on attachment or prosocial behavior within the context of psychopathology (e.g., depression, addictive behaviors, substance abuse; (3) Qualitive research; (4) Research with the main purpose of developing, adapting, and, thus, validating measures of prosocial behavior; (5) Studies that did not accurately or directly assess or measure parental attachment (that did not follow Bowlby’s or Ainsworth’s conceptualization) or prosocial behavior; (6) Papers analyzing intervention programs; (7) Other publications that were not peer-reviewed papers (e.g., books, chapters, conference or poster presentations).

**Table 3 children-09-00874-t003:** Synthesis of the sample dimensions, the participants’ age and ethnicity, the instruments, the results, and the quality of the selected articles.

Articles’ ID, Authors (Date)	N	M Age (SD)	Ethnicity	Attachment Measures	Prosocial Behavior Measures	Results (Associations between Prosocial Behaviors, PB, and Attachment Security, AS)	Q-SSP ^a^ Score
1. Beier et al. (2019) [14]	137 (79 females, 57.66%)	4.32 years (0.50)	Mostly African-American, 66.4%	Preschool Strange Situation procedure (PSS, [17]).	Observation and coding of behaviors such as helping, sharing, and comforting.	AS predicted PB: β = 0.236 * AS predicted helping behaviors: β = 0.651 ** Attachment avoidance predicted helping behaviors: β = −0.759 **	12
2. Bureau & Moss (2010) [40]	129 (69 females, 53.48%)	T1: 6.3 years (1.1)	-	Reunion procedure [41] and Attachment Story Completion Task [42].	Prosocial Behavior Questionnaire [43].	No differences were found in PB levels throughout the different attachment styles (T1: *F* = 1.2; T2: *F* = 0.58, both *p* > 0.05).	11
3. Eceiza et al. (2011) [23]	154 (47% females)	7.39 years	-	Separation Anxiety Test [44,45].	*Profil Socio-Affective* [46].	Ambivalent and secure children showed higher levels of prosocial behavior (F = 5.295 **)	10
4a. Kim & Kochanska (2017) [22]–Family Study	101 (51 females, 50.49%).	T1: 15 months	Mostly Caucasian (80–90%)	Attachment Q-Set (AQS, version 3.0; [28,29]).	Prosocial Behavior scale of HealthBehavior Questionnaire [47].	Direct effect of AS on mothers and PB: β = 0.03, *p* > 0.05 Direct effect of AS to fathers and PB: β = 0.14, *p* < 0.10.	14
4b. Kim & Kochanska [22] (2017)–Play Study	186 (90 females, 48.39%)	T1: 30 months	Mostly Caucasian (70–90%)	AQS, version 3.0; [28,29]).	Infant-Toddler Social and Emotional Assessment [48].	Direct effect of AS on PB: β = 0.08 *;	10
5. Laible et al. (2004) [21]	246 (70% females)	18.6 years (1.61)	15% Caucasian, 13% African-American, 59% Latino	Inventory of Parent and Peer Attachment, IPPA [27].	Global index of prosocial responding [49].	Correlation coefficient: Between parent AS and PB = 0.21 **	12
6. Li et al. (2020) [31]	425 (246 females, 57.88%)	13.97 years (1.67)	Mostly Asian (90–100%)	IPPA-Revised Chinese version [50].	Strengths and DifficultiesQuestionnaire (SDQ [30], Chinese version).	Self-reported PB and mother reported attachment avoidance: β = −0.11 * Self-reported PB and mother reported attachment ambivalence: β = −0.10 * Self-reported PB and self-reported AS: β = 0.32 **	5
7. Marcus & Kramer (2001) [35]	107 (55 females, 51.40%)	64 months	-	Strange Situation, SS [51].	Parent-rating of children social competence [52].	Correlation coefficients: AS and prosocial orientation: 0.57 **Attachment insecurity and prosocial initiative: −0.48 ** AS and prosocial initiative: 0.38 ** Attachment insecurity and prosocial initiative: −0.26 *	14
8. Michiels et al. (2010) [24]	552 (299 females, 54.27%)	11.27 years (0.82)	Mostly Caucasian (92%).	Security Scale (Dutch version: [53]).	SDQ (Dutch version: [54]).	Maternal and paternal AS, individually, were not significant predictors of PB (*t =* 1.357 and *t* = 1.663, respectively, both *p* > 0.05).	13
9. Panfile8 & Laible (2012) [36]	63 (30 females, 47.61%)	36 months	Mostly Caucasian (81%)	Attachment Q-Set version 3 [28,29].	Observation of children’s responses to crying (based on [55]).	Correlation between AS and PB = 0.08, *p* > 0.05, weak and non-significant.	14
10. Profe et al. (2021) [32]	520 (42% females)	12.33 years (0.52)	Mostly mixed-race (46%) and Caucasian (37%)	IPPA [27].	ProsocialTendencies Measure, PTM [56].	Structural equation model coefficients: β Maternal AS and Global PB: 0.04, *p* > 0.05. β AS to Father and Global PB: 0.01, *p* > 0.05. Individual correlations coefficients: Maternal AS and Global PB: 0.10 * AS to father and Global PB: 0.06, *p* > 0.05.	13
11. Shoshani et al. (2021) [37]	1426 (681 females, 47.76%)	11.97 (2.01)	Mostly Jewish (97%)	Attachment Style Classification Questionnaire [57].	SDQ [30].	Correlation between AS and PB: 0.17 *** (positive and significant)	13
12. Simons et al. (2021) [58]	68 (36 females, 52.94%)	13 years, 3 months (4 months)	Mostly Caucasian	IPPA [27].	Prosocial items (teacher and parent report, based on [59,60]).	Maternal and paternal AS were not significantly or positively correlated with PB (self-reported, −0.07, 0.06; parent-reported, −0.11, −0.10; or teacher-reported, −0.21, −0.27).	13
13. Thompson & Gullone (2008) [38]	281 (168Females, 59.78%)	14.83 years (1.71)	-	IPPA-Revised [61]	SDQ [30].	Correlation between PB and AS: 0.25 ***	14
14. Tur-Porcar et al. (2018) [39]	1447 (49.6% females)	9.27 years (1.36)	Mostly Caucasian (79.5%) and Latinos (12.1%)	Security Scale (Spanish version: [62]).	*Escala de conducta prosocial* (Spanish version [63]).	Correlations: between maternal AS and PB: 0.291 ***; between paternal AS and PB: 0.248 ***	12
15. Vagos & Carvalhais (2020) [33]	375 (203 females, 54.1%)	16.62 years(1.03)	-	IPPA (Portuguese version: [64]).	Peer Experience Questionnaire–Revised (Portuguese version: [65]).	Significant structural equation model coefficient: maternal AS and PB, β = 0.017 *	13
16. Zhao et al. (2020) [34]	1177 (51.8% females)	15.37years (1.71)	Mostly Asian (90–100%)	IPPA–Chinese simplified version [66].	PTM [56].	Non-significant structural equation model coefficients: maternal AS and PB, β = 0.01 paternal AS and PB, β = −0.03	12

a. Quality Assessment Checklist for Survey Studies in Psychology (Q-SSP) score; * *p*-value < 0.05; ** *p*-value < 0.01; *** *p*-value < 0.001.

## Data Availability

Not applicable.

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
