# Peer review of "Attachment and the Development of Prosocial Behavior in Children and Adolescents: A Systematic Review"

_children, 2022, doi:10.3390/children9060874_

Round 1

Reviewer 1 Report

Excellent review work on a topic that is relevant and topical. I have not found any improvements to add to your work. Only in the discussion section I would add something that goes beyond what is said.

Regarding the explanations you give for the inconsistency of the results of the studies you have reviewed, I would dare to suggest that you do not forget two possible sets of factors that I succinctly expose:

1) aspects related to the “dark side” of the attachment hormone (oxytocin), that seems to act depending on social cues with pro-social or anti-social effects, and,

2) aspects related to what from Ecological Systems Theory would tell to the exosystem (e.g. neighborhood) and the macrosystem (e.g. attitudes and ideology) of these close developmental contexts.

I think the key question is to whom these prosocial behaviors are directed and what are the facilitating (social) contexts?

I understand that the suggested topics go beyond what is usually asked for in a discussion. Furthermore, I know that in the studies reviewed these topics have not appeared. But I also believe that a systematic review can be the place where a critical stance can be taken, and where and where some challenging proposals can be made to be explored.

Reviewer 2 Report

I think that the total number of items starting from the database search should be added, so that it is possible to perceive which universe is starting from.

 The final number of articles that remains is too small to sustain a bibliometric study of these characteristics.

I think there is a need to collect many more studies on the subject that have appeared in Spain and Italy in recent years and that also have a substratum

Sociocultural very different from the American.

The SES distribution of the samples is surprising, they seem skewed.
